




# Improved Observations of Turbulence Dissipation Rates from Wind Profiling Radars

Katherine McCaffrey[1,2], Laura Bianco[1,2], and James M. Wilczak[2]

[1]University of Colorado, Cooperative Institute for Research in Environmental Sciences at the NOAA Earth System Research Laboratory, Physical Sciences Division, 325 Broadway, Boulder, CO 80305-3337
[2]NOAA Earth System Research Laboratory, Physical Sciences Division, 325 Broadway, Boulder, CO 80305-3337

*Correspondence to:* Katherine McCaffrey (katherine.mccaffrey@noaa.gov)

**Abstract.** Observations of turbulence in the planetary boundary layer are crucial for validation of parameterizations in numerical weather prediction models. However, these observations are sparse. For this reason, demonstrating the ability of commonly-used wind profiling radars (WPRs) to measure turbulence dissipation rates would be greatly beneficial. During the XPIA field campaign at the Boul-

der Atmospheric Observatory, two WPRs operated in an optimized configuration, using high spectral resolution for increased accuracy of Doppler spectral width, specifically chosen to measure turbulence from a vertically-pointing beam only. Multiple post-processing techniques, including different numbers of spectral averages and peak-processing algorithms for calculating spectral moments, were analyzed to determine the most accurate procedures for measuring turbulence dissipation rates us-

ing the information contained in the Doppler spectral width, and compared to sonic anemometers mounted on a 300-meter tower. The optimal settings were determined, producing a constant low bias, which was later corrected. Resulting measurements of turbulence dissipation rates correlated well ($R^2 = 0.57$) with sonic anemometers, and profiles up to 2 km from the 449-MHz WPR and 1 km from the 915-MHz WPR were observed.

## 15  1  Introduction

In the full energy balance of the atmosphere, the turbulent motions are the most difficult to predict, and require parameterizations in numerical models to estimate the contributions at scales smaller than simulations can resolve. Validation of these parameterizations, however, requires observations that are often not available or are limited in time and space. Historically, sonic anemometers mounted

on tall towers or aircraft have provided high-quality *in situ* observations for this purpose, but towers are rare and aircraft are expensive. Wind profiling radars (WPRs) introduced the possibility of observing full profiles of turbulence in the planetary boundary layer (PBL), first introduced by Hocking (1985), with turbulence intensities and dissipation rates being observed using backscatter intensities and Doppler spectral widths. The spectral width method of estimating turbulence dissipation rates





was shown to be the more promising method (Cohn, 1995), and much progress has been made to improve spectral width measurements through the removal of broadening effects (Gossard, 1990; Nastrom, 1997; Nastrom and Eaton, 1997; White et al., 1999). The full integration of the method into routine use has not taken place, however, due to the limitations in measuring the full range of dissipation rates seen by other, high-frequency instrumentation, such as sonic anemometers. Shaw

and LeMone (2003) used a 915-Mhz WPR to measure dissipation rates using spectral widths and found higher values than the *in situ* observations, even after the non-turbulent broadening terms were removed from the observed widths. Here, we use a similar 915-MHz WPR set up with higher spectral resolution for more accurate measurements of spectral widths. Along with several parameters and post-processing techniques, we investigate the ability of the 915-MHz WPR, as well as a 449-

MHz WPR, run adjacent to a highly-instrumented 300-meter tower, to measure accurate turbulence dissipation rates throughout the PBL.

## 2   Observations

All observations in this study were gathered at the Boulder Atmospheric Observatory (BAO), operated by the National Oceanic and Atmospheric Administration's Earth System Research Laboratory.

The site has gently-sloping terrain, and is located 20 km east of the foothills of the Front Range of the Rocky Mountains, about 30 km north of Denver, Colorado. A 300-meter meteorological tower instrumented with sonic anemometers and temperature and humidity sensors is located at the site and, during the eXperimental Planetary Boundary Layer (PBL) Inter-comparison Assessment (XPIA) in March to June of 2015, was instrumented with pairs of sonic anemometers at 6 heights up to 300 m.

Two wind profiling radars were functioning as well, from 1 March to 30 April 2015, located 600 m from the tower at the BAO the Visitors' Center. A detailed description of the field campaign and all instruments included can be found in Lundquist et al. (2016).

### 2.1   Sonic Anemometers

During XPIA, the 300-meter tower at the BAO was equipped with twelve Campbell Scientific

CSAT3 sonic anemometers (commonly called sonics, provided by the Characterizing the Atmospheric Boundary Layer (CABL) program of the National Center for Atmospheric Research and the University of Colorado) mounted on two booms extending to $154°$ and $334°$ from North, at 50 m intervals from 50 to 300 m, sampling at 20 Hz. The sonics recorded three-directional velocities, with **u** aligned into the boom, and **v** aligned 90 degrees to the left. The tilt correction and rotation algorithm

of Wilczak et al. (2001) was applied to these data to remove any vertical tilt of the instrument ($< 2°$ for all instruments), and to align the mean velocities into the 30-minute mean wind direction (so that $\overline{\mathbf{v}} = 0 \text{ m s}^{-1}$). Data in the wake of the tower were rejected, based on the 1-minute mean winds from the upstream boom, from $319 - 344°$ and $121 - 197°$ (from N), as determined by McCaffrey et al.





(2016b). Data were also quality-controlled for signal amplitude, signal lock, and the difference in the

speed of sound between the three non-orthogonal axes. The tilt-corrected, rotated, quality-controlled

data from pairs of sonics were then used for calculations of turbulence dissipation rates as detailed

in Section 3.1, and averaged over 30-min intervals for comparison to the WPR observations.

### 2.2 Wind Profiling Radars

Two wind profiling radars are included in the analysis: a 449-MHz radar and a 915-MHz. For each

hour, the systems ran in standard acquisition mode for 25 minutes for three-dimensional consen-

sus winds, in Radio Acoustic Sounding System (RASS) mode for 5 minutes for measurements of

virtual temperature, and in an optimized turbulence mode with a vertically pointed beam only for

the remaining 30 minutes. Time series of backscatter intensity were saved during the 30 minutes of

turbulence mode to allow the testing of different post-processing settings and methods. The radars

measure the velocity in the radial direction (i.e., vertical velocity in turbulence mode), averaging

across a quasi-cylindrical volume with height, $\Delta R$. The details of the radar set-ups are shown in

Table 1. WPR backscatter intensity time series were filtered for birds, radio-frequency interference,

ground clutter, and other non-atmospheric contamination using wavelet and Gabor post-processing

techniques before computing Fast-Fourier Transformations (FFTs) to obtain Doppler spectra (Jor-

dan et al., 1997; Lehmann, 2012). Spectral width measurements were not affected by the application

of these filters. If desired, averaging of Doppler spectra can be performed to lower the noise level,

thereby increasing the Signal-to-Noise-Ratio (SNR) and lengthening the dwell time (since the dwell

time is a function of the number of spectral averages), and then the first and second spectral moments

(mean velocity and velocity variance, respectively) are obtained through peak-processing algorithms.

A threshold that determines whether a signal is discernible was applied to the spectra based on the

SNR. The threshold, developed by Riddle et al. (2012), defines the minimum SNR for detectability

to be

$$SNR_{min} = 10 log \left[ \frac{25 \left( NSPEC - 2.3125 + \frac{170}{NFFT} \right)^{1/2}}{NFFT \times NSPEC} \right], \tag{1}$$

where $NFFT$ is the number of points used in the fast Fourier transform and $NSPEC$ is the num-

ber of spectral averages. Time series and Doppler spectra were post-processed to obtain velocity

moments and noise levels for analysis. The six levels of the sonic anemometers overlapped with six

of the 915-MHz profiler's range gates (50 to 300 m), and four of the 449-MHz WPR's (150 to 300

m).

### 3 Dissipation Rate Calculations

When modeling the turbulent atmosphere, the budget of the Turbulent Kinetic Energy (TKE) is

crucial to understanding the small-scale processes that models cannot resolve and, therefore, must





be parameterized. The full TKE budget can be described as

$$\frac{D}{Dt}(TKE) + \nabla \cdot T = P - \varepsilon \tag{2}$$

where $\frac{D}{Dt}$ is the material derivative, $T$ is the turbulent transport, $P$ is the production, and $\varepsilon$ is the dis-
sipation rate, lost to heat. The dissipation rate, $\varepsilon$, can be estimated with several methods, as shown
by Kaimal et al. (1968); Gossard (1990); Shaw and LeMone (2003). The foundational work by
Kolmogorov (1941) provided a basic dimensional argument for relating the dissipation rate to the
transfer of kinetic energy between scales of motions, assuming isotropic, homogeneous, stationary
turbulence. In this case, energy is injected at large scales and transferred to small scales, where it dis-
sipates, with no energy lost in the intermediate scales, called the "inertial range." The energy spectral
density for velocity component, $i$, can therefore be represented in terms of only the dissipation rate
and wavenumber (scale), $k$:

$$\phi_{E,i} = \alpha_i \varepsilon^{2/3} k^{-5/3}, \tag{3}$$

where $\alpha_i$ is a constant. The energy spectrum can be written in terms of frequency, $f$, instead of
wavenumber assuming $k = 2\pi f / \overline{u}$.

$$\phi_E(f) = \alpha_i \left(\frac{2\pi}{\overline{u}}\right)^{-5/3} \varepsilon^{2/3} f^{-5/3}. \tag{4}$$

The integral of the energy spectrum in the inertial range is equal to the total variance. Assuming this
basic form for the energy spectrum, and the relationship to $\varepsilon$, the dissipation rates can be calculated
from both the radar profilers and the sonic anemometers, using different methods for each dataset,
to close the full TKE budget.

### 3.1  $\varepsilon$ from Sonic Anemometer Energy Spectra

For all time intervals when $\phi_E(f)$ has a $-5/3$ power law, the observed power spectral density can
be used to calculate the dissipation rate. This can be accomplished using *in situ* sonic anemometers,
which operate at a very high frequency, and capture the inertial range. WPRs do not collect data at a
high enough frequency to observe the inertial range, and therefore must rely on spectral widths for
their small-scale variance measurements (McCaffrey et al., 2016a).

Using the high-sampling frequency measurements by the sonic anemometers, the variance ob-
served can be directly obtained from the energy spectrum for the dissipation rate. The entire spec-
trum, however, contains biases that must be removed before calculating the dissipation rates. To
account for the bias introduced from Taylor's hypothesis used to move from frequency to wavenum-
ber space, Wyngaard and Clifford (1977) obtained a modification, $T_{xx}$, on the observed spectrum as
follows:

$$T_{uu} = 1 - \frac{1}{9}\frac{\sigma_u^2}{\overline{U}^2} + \frac{2}{3}\frac{\sigma_v^2}{\overline{U}^2} + \frac{2}{3}\frac{\sigma_w^2}{\overline{U}^2} \tag{5}$$

$$T_{vv} = T_{ww} = 1 - \frac{1}{9}\frac{\sigma_u^2}{\overline{U}^2} + \frac{2}{3}\frac{\sigma_v^2}{\overline{U}^2} + \frac{1}{3}\frac{\sigma_w^2}{\overline{U}^2}, \tag{6}$$





where the subscripts $x = u$, $v$, and $w$ denote the components of the velocity vector. Similarly, the averaging along the sonic path introduces a bias in the amplitude of the energy spectrum, so Kaimal et al. (1968) developed a spectral transfer function to reduce this impact:

$$T_{stf}(kd) = \frac{sin^2\left(kd/2\right)}{\left(kd/2\right)^2},\qquad(7)$$

where $d$ is the path-length, equal to 15 cm for the sonic anemometers used here. The modified energy
spectrum is then defined as:

$$\phi_{new}(k) = \phi_E(k) \cdot T_{stf}(kd) \cdot T_{xx},\qquad(8)$$

Figure 1 shows one 15-minute velocity spectrum for the horizontal (blue) and vertical (red) velocities with (bright colors) and without (pale colors) the adjustments. It is seen that the adjustment is minimal, but used for thoroughness, since it extends the inertial range. Equation 8 can be inverted and
averaged over the inertial range (here, from $8\mathrm{x}10^{-2}$ to $2\mathrm{x}10^{0}$ $s^{-1}$, as denoted by the solid vertical lines in Fig. 1), as in Lien and D'Asaro (2006), to solve for $\varepsilon$:

$$\varepsilon = \frac{1}{\alpha_u^{3/2}}\left(\frac{2\pi}{\overline{u}}\right)^{5/2}\left\langle\left[f^{5/3}\phi_{new}(f)\right]^{3/2}\right\rangle,\qquad(9)$$

where the angled brackets are averages in frequency. Spectra were computed for each 15-minute interval, with corresponding dissipation rates. The dissipation rates were then averaged over 30 mins
to compare with those obtained from the WPRs.

**3.2   $\varepsilon$ from Radar Spectral Widths**

The method of calculating dissipation rates from WPRs uses the Doppler spectral width, rather than the vertical velocity spectrum. The Doppler spectrum measured by a profiling radar contains both the resolved velocities in the mean wind speed, and the unresolved velocities in the spectral width.
The spectral width of the Doppler spectrum is twice the standard deviation, $\sigma_m$, of the unresolved velocities in the measurement volume during each dwell. This total measured variance, $\sigma_m^2$, includes the effects of turbulence, as well as non-turbulent effects such as wind shear and antenna effects. The total variance is made of independent contributors, so it can be summarized as follows:

$$\sigma_m^2 = \sigma_s^2 + \sigma_t^2,\qquad(10)$$

where $\sigma_s^2$ is the variance due to wind shear and beam broadening effects, and $\sigma_t^2$ is due to turbulence (Gossard, 1990). Nastrom (1997) has determined the shear and beam-broadening term to depend on the mean wind transverse to the beam axis, $V_T$, the mean wind shear, $du/dz$, and the antenna properties, and for a vertically-pointing beam, is determined to be

$$\sigma_s^2 = \frac{\nu^2}{3}\left(V_T^2 + \left(\frac{du}{dz}\right)^2\frac{\Delta R^2}{12}\right)\qquad(11)$$



where $\nu$ is the half-width to the half-power point in the antenna pattern. Nastrom (1997) also de-
termined the broadening contribution due to gravity waves across the radar beam, but for a single,
vertically-pointing beam, the contribution is small, and therefore neglected. In some cases, however,
the contribution to the spectral width from the shear and beam broadening term, $\sigma_s^2$, is larger than
the total measured width, $\sigma_m^2$, creating negative values of dissipation rates. These are not physi-
cally meaningful, so these values are rejected. Despite possibly skewing the 30-minute averages by
neglecting values which may be small (Dehghan et al., 2014), comparisons showed that all other
solutions (e.g. substituting with 0 or a small value of $\sigma$, or not removing any broadening) pro-
duced unphysical or statistically worse results. Figure 2 shows the percentage of 2-minute dwells
($NSPEC = 8$) that experience $\sigma_s^2 > \sigma_m^2$. The 915-MHz WPR experienced this up to 40 % of the
time, though the fraction decreases with more spectral averages, as the measured widths are broad-
ened with longer dwell times (and the beam-broadening remains constant). The 449-MHz, with a
narrower beam and higher SNR, is not as impacted by this issue, except at the lowest level, where
ground clutter causes noisier spectra.

The work by Frisch and Clifford (1974), Labbitt (1981), and Gossard et al. (1998) led White et al.
(1999) to an expression for the relationship between the contribution to the spectral width due to
turbulence and the dissipation rate, in a spectral form:

$$\sigma_t^2 = \frac{\alpha \varepsilon^{2/3}}{4\pi} \int \int \int k^{-11/3} \left[ 1 - \left( \frac{k_1}{k} \right)^2 \right] \left[ 1 - \frac{sin^2\left( k_2 L/2 \right)}{\left( k_2 L/2 \right)^2} \exp\left[ -b^2 k_1^2 - a^2 \left( k_2^2 + k_3^2 \right) \right] \right] dk_1 dk_2 dk_3, \tag{12}$$

where $\alpha = 1.6$ is a Kolmogorov constant, $L = V_T t_D$ ($t_D$ is the dwell time), $a$ is the half-diameter of
the (circular) beam cross-section, $b$ is the half-length of the pulse, and $k_i$ are the three components
of the wavenumber. To solve the integral in Equation 12 requires converting to spherical coordinates
and integrating

$$I = 12\Gamma\left( \frac{2}{3} \right) \int_0^{\frac{\pi}{2}} d\phi \int_0^{\frac{\pi}{2}} d\theta sin^3\theta \left( b^2 cos^2\theta + a^2 sin^2\theta + \frac{L}{12} sin^2\theta cos^2\phi \right)^{1/3}, \tag{13}$$

where $\Gamma$ is the gamma function. This equation can be numerically integrated, and Equation 12 can
be solved for $\varepsilon$.

The dissipation rates were estimated for the 30 minutes of turbulence mode, when the backscatter
intensity time series were collected, and after different post-processing and moments'-calculations
methods were performed.

## 4 Post-Processing Procedures

During the calculation process for spectral moments from WPRs, there are several options and pa-
rameters to be considered, with the possibility of improving the accuracy of the spectral width mea-




surements. These options include radar set-up, backscatter intensity time series filtering, Doppler-spectral processing, and moments' calculations, all of which have an effect on the final spectral width used for dissipation rates. Here we will investigate the effect of standard and multiple peak-processing methods and noise level thresholds of moments' calculations, and spectral averaging on the eddy dissipation rate as determined with the WPRs' spectral width, using the *in situ* observations from sonic anemometers for comparison.

### 4.1 Moments' calculations: Standard vs Multiple Peak-Processing

Once the Doppler spectra are calculated (with wavelet and Gabor filtering applied), and processed for ground clutter (Riddle and Angevine, 1991) and other interference reduction, the first two moments of the Doppler spectra are calculated by one of two common methods. The first is the Standard Peak Processing method, SPP, using the basic method of finding the highest Doppler peak at each range gate, then integrating between the velocities, $\nu_1$ and $\nu_2$, above the noise level (maximum or mean level). The second method of calculating the moments, the Multiple Peak Processing method, MPP, finds up to three peaks at each range gate with the highest power, then uses continuity in time and space (vertical profiles) to determine which peak is most likely to be the true signal. The velocity range of the peak is determined by the crossing of the mean noise level or an increase in power due to an adjacent peak. The velocity limits, $\nu_1$ and $\nu_2$, of the chosen peak then go into the calculation of first and second moments for velocity and spectral width. MPP is the preferred method for the measurement of first-moment winds, as determined in Gaffard et al. (2006), but the impact on the width has not been studied.

When using SPP, the noise level threshold that determines the velocity limits in the calculation of the spectral width can be set to either the maximum noise level of the spectrum, or the mean noise level. The common choice is the maximum noise level since it is the most conservative for removing noise, and produces a more accurate first moment of the spectrum. However, a non-atmospheric signal could create an artificially high maximum noise level, causing the spectral width to be narrowed. The mean noise level in these cases would be more representative of the true noise in the spectrum, and allows the measured spectral widths to be realistic. A comparison of dissipation rates using these three methods with the *in situ* measurements from the sonic anemometers will indicate which moments' calculation method is most accurate for measuring spectral widths and, consequently, dissipation rates. All other variables in the calculation of dissipation rate are equal across different moments' calculation methods, so the accuracy of dissipation rates indicates the accuracy of the spectral width measurements in each method.

### 4.2 Spectral Averaging

Each dwell collected by the 449-MHz WPR spans about 13 seconds (and the 915-MHz, about 17 seconds; see Table 1), capturing only a short period of the atmosphere's motions. Therefore, turbu-



lence observed in that dwell time does not completely capture the full characteristics of the flow. Most turbulence statistics, such as turbulence intensity or turbulent kinetic energy, are calculated using 2- to 30-minute averages of fluctuations to include a more complete range of scales of turbulence. In the case of Doppler spectra from pre-determined radar pulses, multiple dwells can be averaged

together to span a longer time period of fluctuations, resulting in more representative turbulence statistics. However, averaging over periods that are too long, and therefore non-stationary, will result in broadening of the spectral peak that is due to a shifting mean velocity, rather than due to true fluctuations from turbulence. McCaffrey et al. (2016a) analyzed the impact of lengthened averaging times on vertical velocity variance measurements, and found that the optimal time for spectral width

calculations was 2 minutes. Here, an analysis was performed to determine the length of time, set by the number of spectral averages, which produces the most accurate dissipation rates compared to the *in situ* observations from the sonics.

## 5  449-MHz WPR - Sonic Anemometer Comparisons of $\varepsilon$

With the use of the *in situ* sonic anemometer measurements as the baseline, the different post-

processing techniques presented in Sections 4.1 and 4.2 can be analyzed for their impacts on the resulting dissipation rates. Figure 3 summarizes the behavior of comparisons between the 449-MHz WPR and the sonic anemometers at their 4 overlapping heights, using MPP (blue) and SPP, with maximum (red) and mean (green) noise levels, as a function of the different numbers of spectral averages. With the un-averaged spectra having a dwell time of 13 s, using $NSPEC = 42$ produces

dwells of approximately 10 minutes. In the correlation (measured as $R^2$, Fig. 3a), mean absolute error (MAE, Fig. 3b), and fractional bias (Fig. 3c), MPP does significantly worse than both settings of SPP. A scatter plot of dissipation rates from MPP and the sonic anemometer (Fig. 4) shows that, even at the optimal averaging time ($NSPEC = 4$) as determined by Fig. 3, MPP often highly underestimates the spectral width that contributes to the dissipation rate. At this high spectral resolution,

the MPP method identifies multiple peaks within the one, highly-resolved peak, and so computes artificially narrow widths within the true, wider peak. Thus, we conclude that MPP should not be used for calculations of spectral widths, particularly at high spectral resolution.

When using SPP with either the maximum or mean noise level, averaging over more than one dwell is immediately an improvement over the shortest dwell times (Fig. 3). The highest correlation

of $R^2 = 0.57$ occurs with the maximum noise level using $NSPEC = 4$, or about 1-min dwells, and remains constant until $NSPEC = 15$ (Fig. 3a). The mean noise level sees its highest correlation, also of $R^2 = 0.57$, at $NSPEC = 2$ (about $\Delta t = 30$ s), then decreases for longer dwell times. The lowest MAE occurs for both noise thresholds at $NSPEC = 2$, and increases slightly for longer dwells (Fig. 3b). The MAE is lowest at all $NSPEC$ when using the mean noise level. The bias in

SPP is lower using the mean noise threshold at longer time scales, with a minimum (at 100 %) at



$NSPEC = 2$ for the maximum noise level and $NSPEC = 15$ for the mean noise level (Fig. 3c). Optimization of the mean versus maximum noise level is flexible, through the correlation, MAE, and bias. Similar analysis of the vertical velocity variance, as measured by the WPR spectral widths in McCaffrey et al. (2016a), showed the optimal results at $NSPEC = 8$ using the mean noise level in

SPP, and that choice for dissipation rates would also give a minimal bias with small MAE and high correlation, so future analysis herein will use $NSPEC = 8$ and SPP with the mean noise level.

Scatter plots comparing dissipation rates from the sonic anemometers and WPRs using the optimized post-processing procedures are shown in Fig. 5, both as a scatter plot of all 30-min averages (a), and as box-and-whisker plots to more easily see the distribution of values (b). The scatter plot

shows a trend near to 1 for larger sonic anemometer values of dissipation (larger than $10^{-4}$ m$^2$s$^{-3}$), but with a constant offset from the one-to-one line. The scatter at lower sonic-observed values increases, with more radar values that are over-estimated, causing the overall slope of the best-fit line to be much less than 1. The shift in behavior is more visible on the box plot (Fig. 5b), where the distribution of values departs from the trend near $\varepsilon_{sonic} = 10^{-4}$ m$^2$s$^{-3}$, and flattens. At the largest

values, the box plot also shows a flattening, but the scatter plot shows that there are few points in this range, so the statistical significance of the departure from the trend above $\varepsilon_{sonic} = 10^{-1}$ m$^2$s$^{-3}$ is less significant. Between $\varepsilon_{sonic} = 10^{-4}$ and $10^{-1}$ m$^2$s$^{-3}$, the slope of the line fit through the median of each bin (red dashed line in Fig. 5b) has a slope much closer to 1, with the overall low bias seen as well.

## 6   Results from the 915-MHz WPR

The 915-MHz WPR operating during XPIA was set up to have similar temporal and spectral resolution as the 449-MHz, but the different systems produce spectra with different noise levels, and slightly different resolutions (see Table 1). The optimization of dissipation rates through the post-processing techniques of spectral averaging and moments' calculations performed for the 449-MHz

WPR must be completed separately for the 915-MHz WPR. Figure 6 compares the coefficient of determination, $R^2$, MAE, and fractional bias between the 915-MHz WPR and all 6 overlapping heights of sonic anemometers as a function of the number of spectral averages. The correlations are lower, with higher biases overall, for the 915-MHz WPR than the 449-MHz system. Again, MPP is significantly worse at calculating the spectral width (and therefore, dissipation rates). There is less of a

difference between the noise level thresholds for the 915-MHz WPR than the 449-MHz, so for consistency, the mean noise level will be used for further analysis. $NSPEC = 8$ shows the best results in terms of all three quantities shown presented Fig. 6, showing that a time scale of approximately 2 minutes is optimal for measuring spectral widths. Figure 7 uses these settings for computation of dissipation rates, and is presented as a scatter plot (a) and box-and-whisker plot (b), with the median



and 25th-percentiles for each bin of sonic anemometer values. The overall low bias is again evident, and a trend close to 1 is again seen above $10^{-4}$ m$^2$ s$^{-3}$.

## 7 Bias Corrections

The overall biases in dissipation rates seen in Figs. 5a and 7a, and the constant offset in the range $\varepsilon_{sonic} = 10^{-4}$ and $10^{-1}$ m$^2$ s$^{-3}$ in Figs. 5b and 7b suggest that a bias correction could produce more accurate agreement with the sonic anemometers. A bias correction that is a function of the sonic anemometers values cannot be applied since those values, in most cases, will not be available. Using a constant determined by these WPR-sonic comparisons allows a correction to be performed on other datasets without *a priori* knowledge of the true value. Two different methods of correcting the observed bias were tested; one corrects based on a constant determined by the dependence on the sonic anemometers, and one is dependent on the observed radar value. The methods are defined in Table 2 with their respective corrections using a function, $\hat{F}(\varepsilon)$ that is averaged in each case. The biases were calculated separately for the two WPRs for the first month of observations (March), reserving the second month (April) for testing of the corrections.

Figure 8 presents the fractional biases in the 915-MHz WPR as a function of the sonic anemometers ($\varepsilon_{radar}/\varepsilon_{sonic}$, Fig. 8a) and the WPR ($\varepsilon_{sonic}/\varepsilon_{radar}$, Fig. 8b). The correction based on the sonic-dependent bias (Fig. 8a) must be a constant in order to be applied to other datasets, and therefore, an average in the densest part of the range was chosen (from $10^{-3}$ to $10^{-2}$ m$^2$ s$^{-3}$), and is shown as the dashed line. This constant, $c_1 = 0.339$, can then be used to get a corrected value of $\hat{\varepsilon}_{radar}$ as shown in Table 2.

The second method uses a bias correction dependent on the WPR values, rather than the sonic, as in the first method. The WPR-dependent bias (Fig. 8b) has constant behavior over most dissipation values, so the average between $10^{-4}$ and $10^{-2}$ m$^2$ s$^{-3}$ was used (dashed line). Again, the bias-correction factor, $c_2 = 2.784$ is multiplied by each $\varepsilon_{radar}$ value to get the corrected $\hat{\varepsilon}_{radar}$.

To determine the dissipation rates that are most impacted by the bias corrections, Fig. 9 shows the biases in the 915-MHz WPR before corrections (blue line) and after correcting, using the sonic- (red) and radar-dependent (purple) corrections. The observations were binned by the instrument on the dependent axes ($\varepsilon_{sonic}$ in Fig. 9a and $\varepsilon_{radar}$ in Fig. 9b), and averaged, and the fractional bias was calculated for each bin. In Fig. 9a, the ability of the WPR to measure each sonic value of $\varepsilon$ is indicated by the biases, highlighting where the radar over- and under-estimates the dissipation rate. The improvements can be seen above $\varepsilon_{sonic} = 6 \times 10^{-4}$ m$^2$ s$^{-3}$, where the biases in both corrected datasets (red and purple lines) are closer to the 100 % line than the original (blue line). However, below $\varepsilon_{sonic} = 6 \times 10^{-4}$ m$^2$ s$^{-3}$, the constant multiplicative adjustment acts in the wrong direction, and dissipation rates that were originally over-estimated by the WPR are further over-estimated.



Analyzing the average biases as a function of the WPR-measured dissipation rate gives insight
into the accuracy of those measurements (Fig. 9b). The original data set always lies above the 100
% line (blue line), indicating that the true sonic value is nearly always higher than the observed
WPR values, except at the highest values (which are few, as seen in Fig. 8b). The corrections based
on both the radar and the sonic anemometers nearly remove the entire bias for dissipation rates
in the range between $\varepsilon_{radar} = 2\text{x}10^{-5}$ and $10^{-2}$ m$^2$ s$^{-3}$, and improve all measurements below
$\varepsilon_{radar} = 10^{-2}$ m$^2$ s$^{-3}$.

Though the two corrections, based on each instrument, are not mathematically identical, their constant corrections are nearly the reciprocal of one another. This creates bias corrections that are nearly
equal, but since the corrections are defined by an average, and applied to individual points before
further averaging in Fig. 9, the results are not identical. The fact that they are so close, however,
indicates that either correction method can be used.

Applying the same correction methods to the 449-MHz WPR results in similarly improved dissipation rates, using $c_1 = 0.144$ and $c_2 = 5.114$, as seen in the fractional biases in Fig. 10. The bias
correction makes larger improvements as a function of $\varepsilon_{sonic}$ (Fig. 10a), with more accurate dissipation rates measured down to $\varepsilon_{sonic} = 10^{-4}$ m$^2$ s$^{-3}$. As a function of the WPR-measured dissipation
rate, the improvement is significant, removing all biases below $\varepsilon_{radar} = 10^{-2}$ m$^2$ s$^{-3}$. There is no
increase in the bias at the lowest dissipation rates, as was seen in the 915-MHz WPR (Fig. 9b),
showing that the 449-MHz system is more accurate at these levels.

These bias corrections were created based on one month of data (March), and show large improvements. However, the applicability of the corrections can be seen by applying the respective
bias corrections for each radar to the observations taken during April. Figure 11 shows the remaining biases in the dissipation rates from the April datasets, using the sonic-dependent corrections found for the month of March. Both datasets show vast improvements, particularly above
$\varepsilon_{sonic} = 3\text{x}10^{-4}$ m$^2$ s$^{-3}$. In Fig. 12, the sonic-dependent correction constants were used on each
WPR respectively to adjust the entire two months of the original datasets (blue) to obtain the corrected data (red). The biases has been removed in the intermediate-to-large dissipation rates, and
the scatter now falls on the one-to-one line, with MAE $= 0.008$ m$^2$ s$^{-3}$ for the 915-MHz WPR and
MAE $= 0.007$ m$^2$ s$^{-3}$ for the 449-MHz WPR.

When investigating the accuracy of the bias-corrected WPR measurements, box-and-whisker plots
were made (Fig. 13), binned by $\varepsilon_{radar}$, to see the distribution of true dissipation rates (from sonic
anemometers) for each WPR observation. With the bias correction applied, on average (red dashed
line), the radar now accurately matches the sonic anemometer values. For the more-accurate 449-
MHz WPR, only the highest measured $\varepsilon_{radar}$ values do not fall on the one-to-one line, but, as
noted previously, these are infrequent observations. The smallest dissipations rates measured by the
915-MHz WPR show more uncertainty, with broader ranges and medians that do not fall on the

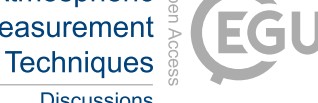

one-to-one line, showing that a measured value in this range is most likely to be underestimating the actual dissipation rate.

An example time series of bias-corrected dissipation rates near 200 m from both WPRs and sonic anemometers is shown in Fig. 14. The diurnal cycle is evident, and is consistently captured by both WPR systems. Several instances of sharp increases in dissipation rate (such as at the beginning of

day 97) are matched well by the WPRs, highlighting the ability of the WPR in capturing turbulent events in the PBL. Full profiles of the dissipation rates are presented in Fig. 15. The growth of the PBL is visible in both datasets, with lower values during nighttimes and increasing dissipation rates in the morning hours of each day. The 449-MHz WPR, with its higher SNR, is able to observe more consistent profiles up to 2 km, while the 915-MHz WPR is limited by its lower SNR, and only when

the spectral widths (and subsequent dissipation rates) are large enough, are the observations possible. This occurs most consistently in the bottom 1 km of the PBL, but occasionally up to 1500 m.

## 8  Conclusions

Using an optimized set-up of two WPRs during the XPIA field campaign in March and April 2015, turbulence dissipation rates were calculated and compared to *in situ* observations from sonic

anemometers on the 300-m tower at the BAO. Using only the vertically-pointing beam and a large number of FFT points to obtain the Doppler spectra with high spectral resolution, post-processing methods were compared to determined the optimal method of measuring dissipation rates from the WPRs. The multiple peak processing (MPP) method of calculating spectral moments showed inaccurate results, often measuring spectral widths that were far too small, most likely due to the high

spectral resolution in this set-up. Using the maximum or mean noise level with the standard peak processing (SPP) method showed small differences, but ultimately the mean noise level was chosen since it produced lower biases in dissipation rates than the maximum noise level. Analysis of the dwell time, dependent on the number of spectral averages, showed that, for both the 915-MHz and 449-MHz WPRs, dwell times of approximately 2 minutes ($NSPEC = 8$) produced the most ac-

curate dissipation rates, supporting the similar results of McCaffrey et al. (2016a) based on vertical velocity variance from spectral widths.

A simple bias correction was applied to the WPR dissipation rates, based on the fractional bias between the radar-measured and sonic-measured dissipation rates. A slightly smaller correction was needed for the 915-MHz WPR, and the constant correction produced improved dissipation rates

above values of $\varepsilon_{sonic} = 6\text{x}10^{-4}$ m$^2$ s$^{-3}$. For the 449-MHz WPR, the full range of values of dissipations rates were improved through a similar constant bias correction. With the bias corrections applied, time series of the dissipation rates from the two WPRs compared very well (especially the 449-MHz WPR) with sonic anemometers, with the entire range of dissipation rates captured throughout the diurnal cycle. The bias corrections were determined for each radar based on one month of



data, with a second month used to test the applicability of the corrections. Other datasets could pro-
vide additional validation of usefulness of the correction for other radar systems or different times
of the year, but the results herein are encouraging. High vertical-resolution profiles of dissipation
rates up to 2 km are obtainable from the 449-MHz WPR and often up to 1 km from the 915-MHz
WPR. These observations will be very useful for the validation of boundary layer parameterizations
in numerical weather prediction models.

*Author contributions.* K. McCaffrey completed the primary analysis with the aid of L. Bianco, and J. Wilczak.
K. McCaffrey prepared the manuscript with contributions from all co-authors.

*Acknowledgements.* Thanks are due to Timothy Coleman, Paul Johnston, Dave Carter for their role in data
acquisition and post-processing. KM was funded by the NRC Postdoctoral Fellowship. The XPIA field pro-
gram was funded under the US Department of Energy's Atmospheres to Electrons (A2e) program and by
NOAA/ESRL. We would like to acknowledge operational, technical and scientific support for the sonic anemom-
etry provided by NCAR's Earth Observing Laboratory, sponsored by the National Science Foundation.



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





| Radar freq (MHz) | 449 | 915 |
|---|---|---|
| Inter-pulse period ($\mu$s) | 33 | 45 |
| No. Coherent Integ. | 24 | 182 |
| $NSPEC$ | 1 | 1 |
| $NFFT$ | 16384 | 2048 |
| First gate height (m) | 154 | 76 |
| # Range gates | 80 | 72 |
| Range gate height (m) | 26 | 25 |
| Dwell time (s) | 12.98 | 16.77 |
| Spectral Resolution (m s$^{-1}$) | 0.025 | 0.01 |

**Table 1.** Radar parameters for the 449-MHz and 915-MHz wind profiling radars, running in turbulence mode for minutes $25 - 55$ of each hour during XPIA from 1 March to 30 April 2015.

| | Sonic-Dependent | Radar-Dependent |
|---|---|---|
| Definition | $\frac{\varepsilon_{radar}}{\varepsilon_{sonic}} = f(\varepsilon_{sonic}) \approx c_1$ | $\frac{\varepsilon_{sonic}}{\varepsilon_{radar}} = h(\varepsilon_{radar}) \approx c_2$ |
| Correction | $\hat{\varepsilon}_{radar} = \frac{\varepsilon_{radar}}{c_1}$ | $\hat{\varepsilon}_{radar} = c_2 \varepsilon_{radar}$ |

**Table 2.** The two bias correction methods, with their definitions and equations of corrections to the observed $\varepsilon_{radar}$ values.

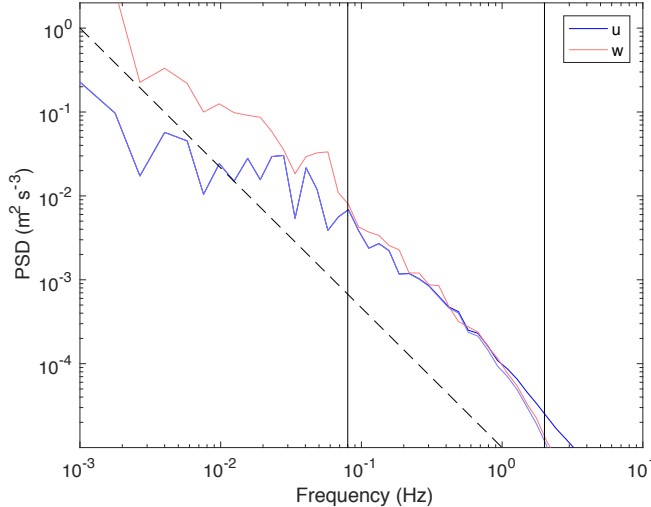

**Figure 1.** Velocity spectra for the horizontal (blue) and vertical (red) velocities with (bright colors) and without (pale colors) the adjustments, for the northwest sonic anemometer at 150 m on the BAO tower. A spectral slope of $-5/3$ is shown as the dashed line for reference, and vertical lines denote the range used for integration.





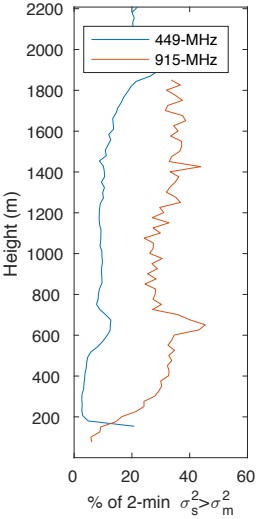

**Figure 2.** Percentage of 2-min dwells ($NSPEC = 8$) when $\sigma_s^2 > \sigma_m^2$ for the 449- (blue) and 915-MHz (red) WPR, using the standard peak-processing method of calculating $\sigma_m^2$.

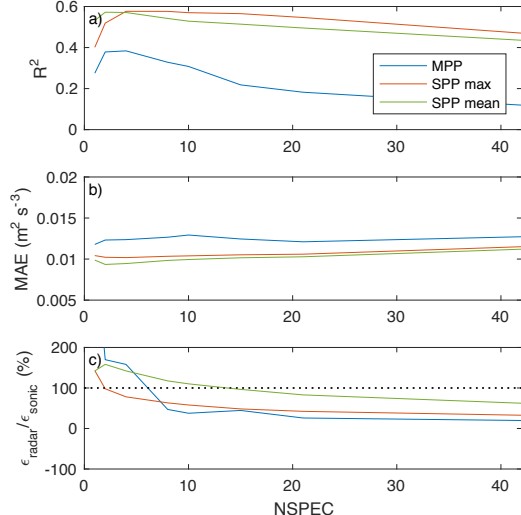

**Figure 3.** a) Coefficient of determination, $R^2$, b) mean absolute error, MAE, and c) fractional bias ($\varepsilon_{radar}$ divided by $\varepsilon_{sonic}$) in dissipation rate between the sonic anemometers and the 449-MHz using MPP (blue) and SPP with the maximum (red) and mean (green) noise level thresholds, as a function of the numbers of spectral averages, NSPEC.





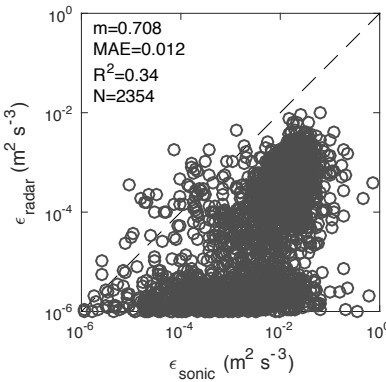

**Figure 4.** Dissipation rates from the sonic anemometer, $\varepsilon_{sonic}$, and WPR spectral widths ( $\varepsilon_{radar}$) from the 449-MHz WPR, using $NSPEC = 4$ and MPP to calculate the spectral moments.

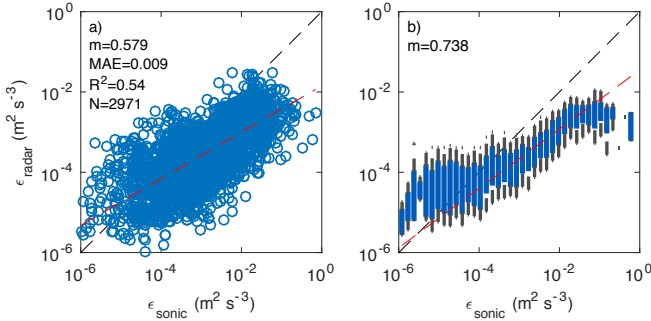

**Figure 5.** Dissipation rates from the sonic anemometer, $\varepsilon_{sonic}$, and WPR spectral widths, $\varepsilon_{radar}$, from the 449-MHz WPR, using SPP with the mean noise level threshold, and $NSPEC = 8$, shown in a scatter plot (a), with slope (m), MAE, $R^2$, and number of points plotted (N) labeled, and as box-and-whiskers (b), with the blue boxes extending to the 25th and 75th percentiles, black lines to the 1st and 99th percentiles, and central lines at the median. The red dashed lines denotes the best fit line to all points in a), and the medians of each bin between $\varepsilon_{sonic} = 10^{-4}$ and $10^{-1}$ m$^2$ s$^{-3}$ in b). The one-to-one line is shown in black dashes on both panels.





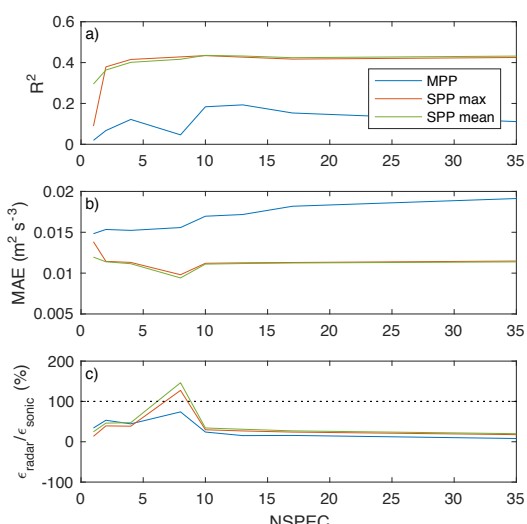

**Figure 6.** Same as Fig. 3, but for the 915-MHz WPR.

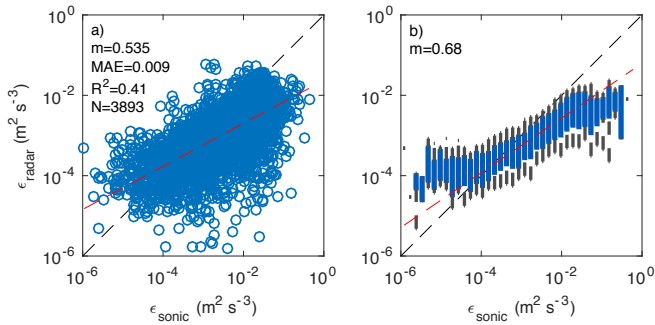

**Figure 7.** Same as Fig. 5, but for the 915-MHz WPR with NSPEC=8 and SPP with the mean noise level.





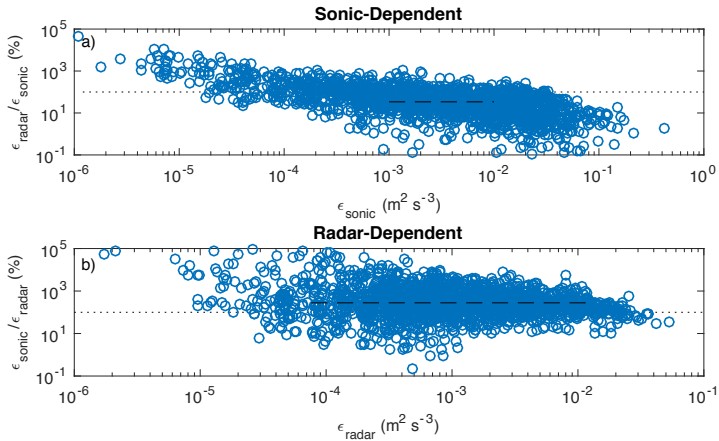

**Figure 8.** Fractional biases, as a percent, in dissipation rate from all 6 heights of the 915-MHz WPR: $\varepsilon_{radar}/\varepsilon_{sonic}$ versus $\varepsilon_{sonic}$ (a) and $\varepsilon_{sonic}/\varepsilon_{radar}$ versus $\varepsilon_{radar}$ (b). Black dashed lines are the averages used in the bias corrections in Table 2.

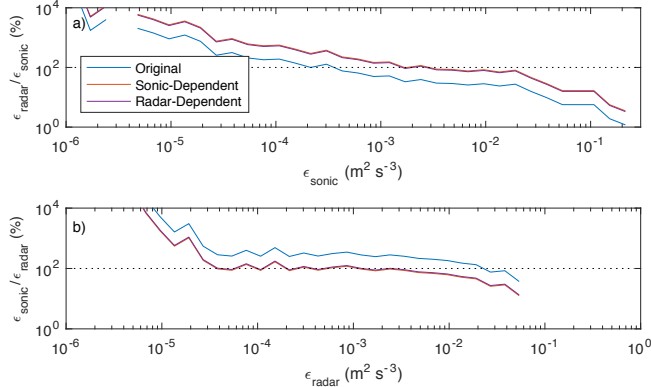

**Figure 9.** Fractional bias in dissipation rate, defined as (a) $\varepsilon_{radar}/\varepsilon_{sonic}$ versus $\varepsilon_{sonic}$ and (b) $\varepsilon_{sonic}/\varepsilon_{radar}$ versus $\varepsilon_{radar}$, during March 2015 from the 915-MHz WPR with no corrections (blue), the sonic-dependent correction (red), and radar-dependent correction (purple).





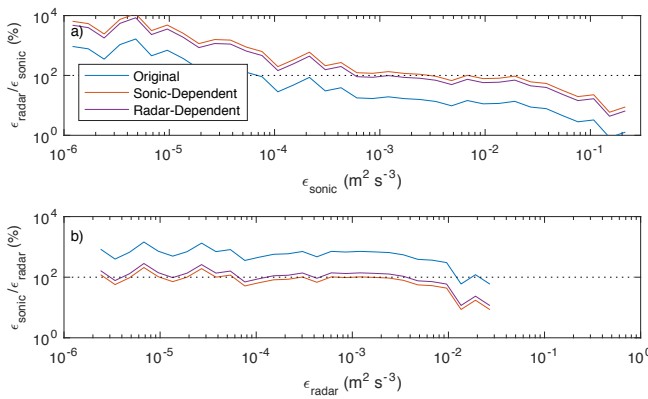

**Figure 10.** Same as Fig. 9, but for the 449-MHz WPR.

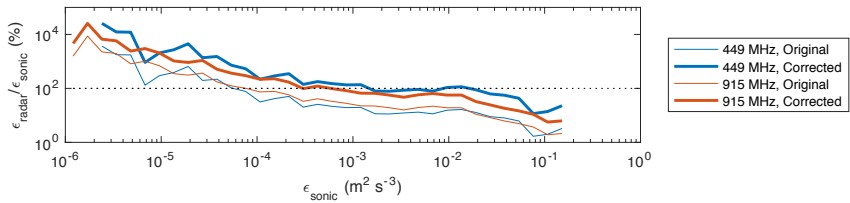

**Figure 11.** Fractional bias in dissipation rate (defined as $\varepsilon_{radar}/\varepsilon_{sonic}$) as a function of $\varepsilon_{sonic}$ during April 2015 from the 449-MHz (blue) and 915-MHz (red) WPRs before applying the bias corrections (thin lines) and after, using the sonic-dependent constant correction found from the month of March (thick lines).

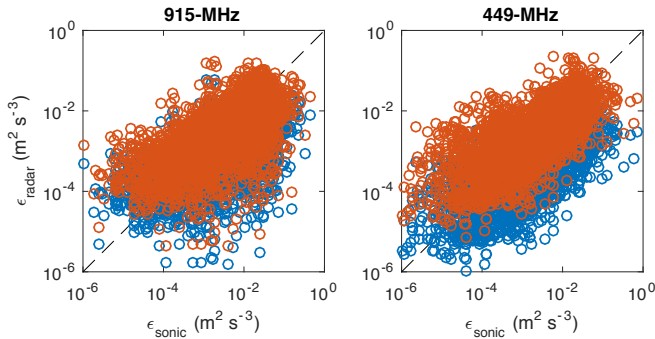

**Figure 12.** Observed dissipation rates, $\varepsilon$, from March and April from the 915-MHz (a) and 449-MHz (b) WPRs versus the sonic anemometers, before (blue) and after (red) bias corrections using the sonic-dependent constant corrections. The black dashed lines are the one-to-one lines.





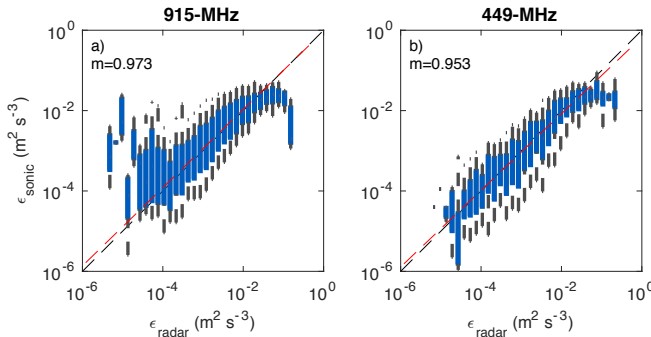

**Figure 13.** Box-and-whisker plots of observed dissipation rates, $\varepsilon$, of the sonic anemometers binned by the 915-MHz (a) and 449-MHz (b) WPR dissipation after using the sonic-dependent constant bias correction. Blue bars extend over the 25th to 75th percentiles, with dashed lines extending to the 90th percentiles. Red dashed lines are fit to the medians of each bin between $10^{-4}$ and $10^{-1}$ $m^2$ $s^{-3}$, with slope, m, labeled. The black dashed lines are the one-to-one lines.

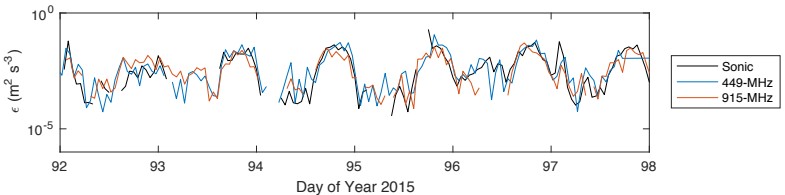

**Figure 14.** Dissipation rate, $\varepsilon$, from 2 - 8 April 2015, from the sonic anemometer (black) at 200 m and the 206 m range gate of the 449-MHz WPR (blue) and the 201 m range gate of the 915-MHz WPR (red) after bias corrections have been applied.





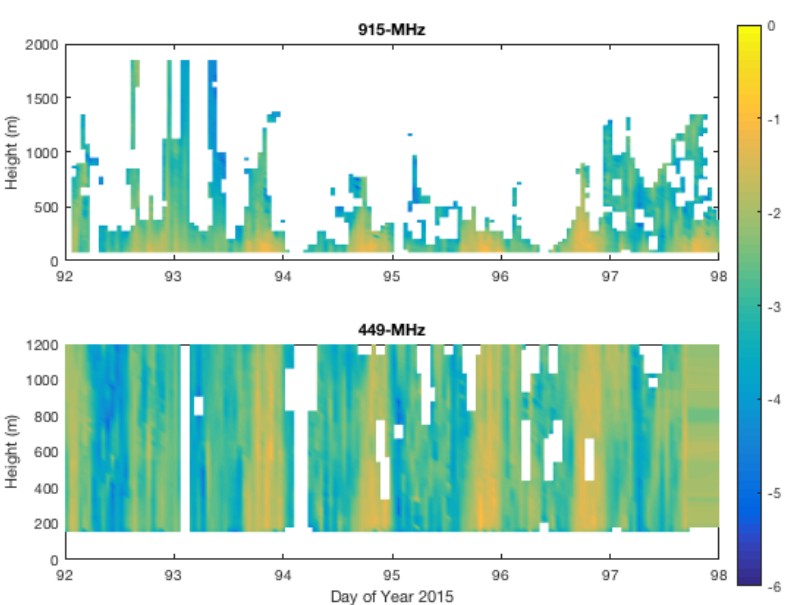

**Figure 15.** Dissipation rate, $\varepsilon$, from 2 - 8 April 2015, from the 915-MHz (top) and 449-MHz WPRs (bottom). Height is AGL.