# Peer review of "Improved Observations of Turbulence Dissipation Rates from Wind Profiling Radars"

_Atmospheric Measurement Techniques, 2016_

## Referee Comment (RC1) · Anonymous Referee #1 · 8 Nov 2016

**1  General comments**

In this article, the authors attempt to measure the dissipation rate of turbulence in the atmospheric boundary layer using vertically oriented, high spectral resolution, wind-profiling radars (WPRs). They compare their measurements with a collocated array of sonic anemometers, and they find a degree of comparison. For the most part, I found this paper clear, easy to read, and interesting.  While I don't share the enthusiasm of the authors in their conclusion, I do think that the techniques and the results presented contribute to scientific progress.  I recommend this article for publication, with the condition that the authors address the scientific and technical questions I've listed below.

**2 Specific comments**

In the first paragraph of Section 3.1, an important point is made on how dissipation rate can be measured by a WPR. However, the reference is to another article, under review, by the lead author of this article. Is there not a more authoritative reference, perhaps Hocking (1985) or Cohn (1995)? Further, this sentence would be a better fit in Section 3.2. References to McCaffrey *et al.* (2016a) are again used in Section 4.2, in Section 5, and in the Conclusions. Might a more canonical reference be appropriate in these locations, as well?

In the third panel of Figs 3 and 6, the lower bound of the $y$-axis should be zero, since the fractional bias values are never less than zero. In the current Figures, this causes the bias values to appear closer to unity than they actually are. Further (and we can argue about this, because perhaps I'm being too picky), in Figs 3, 6, and 8-11, the fractional biases should be plotted as a ratio, not as a percent. Plotting as a ratio would decrease chartjunk. Moreover, plotting as a ratio would make it clearer just how different the $\epsilon$ values calculated from the different sources can be. For example, in Fig 11, the 449-MHz WPR measures bias-corrected $\epsilon$ values that are remarkably close to those captured by the sonics, but only over a range of $[3 \times 10^{-4}, 6 \times 10^{-2}]\,\mathrm{m^2s^{-3}}$. Outside of this range, the values differ by a factor of two to a factor of 100. (An aside: an explanation is offered for why the WPRs don't match the sonics for these large and small values of $\epsilon$ at the end of Section 5. Perhaps this could be expanded and included in the Conclusions?)

Figs 12 and 14 would also benefit from additional detail. I'd like to see $R^2$ values for the plots in Fig 12. They look like blobs. In Fig 14, a subplot of residuals would give a more accurate view of the differences between the dissipation rates measured by the sonics versus those measured by the WPRs. By eye, the plots look close, but the residuals might show otherwise.

I also have a short list of secondary scientific comments. Line numbers are given

where appropriate.

- 71: It would be helpful to move the description of $\Delta R$ to Eqn 11 (I had to flip back to find it)

- In Eqn 8, $\phi_E(k)$ isn't defined; is this the 3D spectrum?

- In Figure 1, it's difficult to distinguish the pale and bright colors

- 135: I suggest "interval" instead of "inertial range", since you can't be sure that this is the inertial range

- 136: I would prefer a more authoritative reference than the one given

- In Figs 9 and 10, the purple is really close to the red; could a different color be used?

- 392: Strike "very"

**3  Technical corrections**

There are several run-on and clumsily structured sentences. One more read-through by the authors, which will likely come about through the review process, will undoubtedly help.

- 21 "Wind profiling radars...": Run-on sentence, with "introduced" used twice

- 41 "A 300-meter...": Run-on sentence

- 44: "6" should be "six"

- 45: "wind profiling radars" should be "WPRs"

- 46: Strike "the" after "BAO"

- 76 "If desired...": Run-on sentence

- In Eqn 3, the subscript $i$ is used to represent velocity components, while on line 121, $xx$ is used

- In Eqns 4, 6, and 11, three different symbols are used for mean wind speed

- 114: The end of the sentence might read better as "which operate at sufficiently high frequencies to resolve the inertial range"

- 120: Strike "used to move from frequency to wavenumber space"

- 133: Strike "It is seen that"

- 151: Run-on sentence

- 164, 255: Use "percent" instead of symbol

- 175: Sentence might read better as "The integral in Equation 12 can be solved by converting to spherical coordinates"
* * *

---

## Referee Comment (RC2) · Anonymous Referee #2 · 14 Dec 2016

**Manuscript ID AMT-2016-322**

Title: Improved observations of turbulence dissipation rates from wind profiling radars

Authors: K. McCaffrey, L. Bianco, and J. M. Wilczak

**General comments:**

The authors present and discuss observations of the turbulent kinetic energy dissipation rate  $\varepsilon$  retrieved from collocated sonic and clear-air wind profiling radar (WPRs) measurements. While the measurements were collected with state-of-the-art instrumentation, the authors appear to not have taken full advantage of the potential of their data. In particular, the authors rely on spectra estimated from unnecessarily long time series, thereby averaging out most of the  $\varepsilon$  variability that is characteristic of the intermittent atmospheric boundary layer. Moreover, the authors do not present any raw data (sonic time series, WPR signal time series) or any conclusive intermediate steps in the data processing, such as Doppler spectra and dwell-time-by-dwell-time time series of spectral-width estimates. So there is no way to find out why the  $\varepsilon$  correlograms (figures 4, 5, and 7) show such excessive discrepancies between the sonic retrievals and the WPR retrievals of  $\varepsilon$  (about three orders of magnitude).

The abstract does not contain any quantitative and hard information that would substantiate the claim in the title ("Improved observations ..."). The introduction lists a number of relevant articles but says almost nothing about the underlying physics. I would have expected the introduction to contain a critical discussion of the strengths and weaknesses of the various techniques to retrieve energy dissipation rates from high-resolution point measurements (sonics) and from radar windprofiler measurements. (Note that the paper does not even have a dedicated discussion section.)

I cannot recommend this paper for publication. The following specific comments may help the authors to rethink their approach and to re-process the data.

**Specific comments:**

- 1. 5f.: ". . . two WPRs operated in an optimized configuration, using high spectral resolution for increased accuracy of Doppler spectral width . . ." The spectral resolution of a Doppler spectrum is determined by the dwell time (and not, as one might erroneously believe, by the pulse repetition period or by the sampling rate or by the coherent integration time). So a high spectral resolution requires a long dwell time. On the other hand, the dwell time should not be excessively long because the dwell time determines also the achievable time resolution. Because the energy dissipation rate  $\varepsilon$  in the atmospheric boundary layer is a highly intermittent quantity, one might want to limit the dwell time such that "coherent structures" can be captured. So in what sense is the WRP operation mode optimized here? What is the "optimum" compromise between good spectral resolution and good time resolution?
- 2. 12f.: "Resulting measurements of turbulence dissipation rates correlated well ( $R^2 = 0.57$ ) with sonic anemometers . . ." — Is this the correlation coefficient between WRP- and sonicretrieved estimates of  $\varepsilon$ , or the correlation coefficient between WRP- and sonic-retrieved estimates of the logarithm of  $\varepsilon$ ? The figures 4, 5, and 7 appear to indicate the latter. A moderate correlation coefficient between the logarithms of  $\varepsilon$ -estimates that vary over five orders of magnitude does not speak in favor of the quality of the retrievals. Please explain.

- 3. 21f.: "Wind profiling radars (WPRs) introduced the possibility of observing full profiles of turbulence in the planetary boundary layer . . ." What do you mean by "full profiles of turbulence"? First, what is a "full" profile? Second, turbulence is characterized by many different parameters, such as the energy dissipation rate, the temperature structure parameter, the inner scale, the temperature variance dissipation rate, the sensible heat flux, the vertical-velocity variance, etc. So what do you mean with "turbulence"?
- 4. 90-95: The TKE budget equation is not relevant for the retrieval of  $\varepsilon$  from sonic or WPR measurements discussed in this paper, and therefore should be deleted.
- 5. 104: ". . . where  $\alpha_i$  is a constant." Do you mean to say that there are three different Kolmogorov constants, one for each velocity component? This is misleading. There are only two different Kolmogorov constants: one for the longitudinal velocity component (the velocity component parallel to the direction of the wave vector) and one for the transverse (lateral) velocity component. The underlying physics can be found in Monin and Yaglom (1975, p. 355).
- 6. 115: "WPRs do not collect data at a high enough frequency to observe the inertial range, and therefore must rely on spectral widths for their small-scale variance measurements . . ." According to Table 1, the WPRs' interpulse periods are 33 and 45  $\mu$ s. Because no coherent integration is performed, this means that the sampling period of the complex WPR signal is higher than 20 kHz, about 1000 times the sonics' sampling rate. So it is definitely not the sampling rate that matters here. Please explain!
- 7. 117f. "Using the high-sampling frequency measurements by the sonic anemometers, the variance observed can be directly obtained from the energy spectrum for the dissipation rate." — It is not the "variance observed" that is used for the retrieval of  $\varepsilon$ . Please clarify!
- 8. 138: "Spectra were computed for each 15-minute interval, with corresponding dissipation rates. The dissipation rates were then averaged over 30 mins to compare with those obtained from the WPRs." — Why do you average over such long periods? It should be possible to get clean  $\varepsilon$  estimates from the sonic data and from the WPR spectral widths for periods of 1 min or even shorter! At a sonic sampling rate of 20 Hz, you have 1200 velocity data points per minute, which is more than enough to provide a clean estimate for  $\varepsilon$ . The same should be the case for  $\varepsilon$ -estimates retrieved from the Doppler spectral widths obtained from 1-min long WPR signal time series.
- 9. 145: "The spectral width of the Doppler spectrum is twice the standard deviation,  $\sigma_m$ , of the unresolved velocities in the measurement volume during each dwell." Whether or not this statement is correct depends on what exactly you mean by "unresolved velocities in the measurement volume." Moreover, the width of the Doppler spectrum has the unit Hz while the unit of  $\sigma_m$  is ms-1, so the ratio between spectral width and  $\sigma_m$  is not a dimensionless number. Please clarify!
- 10. 180: "The dissipation rates were estimated for the 30 minutes of turbulence mode, when the backscatter intensity time series were collected . . ." The dissipation rates are not retrieved from intensity time series but from time series of amplitudes and phases (i.e., from time series of the complex radar signal). Please clarify!
- 11. 186: ". . . backscatter intensity time series filtering . . ." See previous comment. Please clarify!

12. 291: "Each dwell collected by the 449-MHz WPR spans about 13 seconds (and the 915-MHz, about 17 seconds; see Table 1), capturing only a short period of the atmosphere's motions." — For short dwell times, the Doppler spectral width provides information about the spatial variability of the radial wind velocity  $v_r$  within the radar's resolution volume. For long dwell times, the spectral width is contaminated by temporal variations of the mean (averaged over the radar's resolution volume)  $v_r$ . It is the spatial variability of  $v_r$  within the radar's resolution volume that provides the most direct information of  $\varepsilon$ . The fact that  $\varepsilon$ -estimates vary erratically from dwell time to dwell time must not be misinterpreted as instrumental noise that has to be averaged out; more likely, the "noise" represents the intermittent nature of  $\varepsilon$  in the high-Reynolds number ABL (Kolmogorov, 1962; Obukhov, 1962). The local energy dissipation rate in high-Reynolds number turbulence (as in the ABL) is approximately lognormally distributed, such that variations of  $\varepsilon$  over many orders of magnitude within short time scales (i.e., minutes or less) are to be expected. It would be very interesting to see to what extent the collocated sonic and WPR  $\varepsilon$  retrievals track each other at time scales between 10 s and 10 min, rather than at time scales larger than 10 min.

**Bibliography**

- Kolmogorov, A. N., 1962: A refinement of previous hypotheses concerning the local structure of turbulence in a viscous incompressible fluid at high Reynolds number. J. Fluid Mech., 13, 82–85.
- Monin, A. S. and A. M. Yaglom, 1975: *Statistical fluid mechanics Volume 2*. The MIT Press, Cambridge, Massachusetts, 874 pp.

Obukhov, A. M., 1962: Some specific features of atmospheric turbulence. J. Fluid Mech., 13, 77–81.

---

## Author Comment (AC1) · 12 Jan 2017

We thank the reviewer for their detailed reading and critique of our manuscript. We have addressed their concerns, and hope they find this work acceptable for publication!

(1) Referee Comments (2) Author Response (3) Change in manuscript

1 General comments In this article, the authors attempt to measure the dissipation rate of turbulence in the atmospheric boundary layer using vertically oriented, high spectral resolution, wind profiling radars (WPRs). They compare their measurements with a collocated array of sonic anemometers, and they find a degree of comparison. For the most part, I found this paper clear, easy to read, and interesting. While I don't share the enthusiasm of the authors in their conclusion, I do think that the techniques and the results presented contribute to scientific progress. I recommend this article

for publication, with the condition that the authors address the scientific and technical questions I've listed below.

2 Specific comments (1) In the first paragraph of Section 3.1, an important point is made on how dissipation rate can be measured by a WPR. However, the reference is to another article, under review, by the lead author of this article. Is there not a more authoritative reference, perhaps Hocking (1985) or Cohn (1995)? Further, this sentence would be a better fit in Section 3.2. (2) We included a previous, better-known reference for the different scales of turbulence measurements from WPRs in Angevine et al 1994. This sentence was removed from section 3.1, and the reference was made after the second sentence in Section 3.2. (3) line 119: "inertial range. Using the..." line 152: "spectral width (Angevine et al., 1994; McCaffrey et al., 2016)."

(1) References to McCaffrey et al. (2016) are again used in Section 4.2, in Section 5, and in the Conclusions. Might a more canonical reference be appropriate in these locations, as well? (2) The results mentioned here are unique to the McCaffrey (2016) manuscript. It received reviews of minor revisions, and is expected to be accepted shortly, and will likely be published before this one.

(1) In the third panel of Figs 3 and 6, the lower bound of the y-axis should be zero, since the fractional bias values are never less than zero. In the current Figures, this causes the bias values to appear closer to unity than they actually are. (2) Yes, you are right. This has been fixed. (3) Updated (new) Figs 4c and 7c.

(1) Further (and we can argue about this, because perhaps I'm being too picky), in Figs 3, 6, and 8-11, the fractional biases should be plotted as a ratio, not as a percent. Plotting as a ratio would decrease chart junk. Moreover, plotting as a ratio would make it clearer just how different the values calculated from the different sources can be. For example, in Fig 11, the 449-MHz WPR measures bias-corrected values that are remarkably close to those captured by the sonics, but only over a range of [ $3 \times 10-4$ ,  $6 \times 10-2$ ] m2 s -3. Outside of this range, the values differ by a factor of two to a
factor of 100. (An aside: an explanation is offered for why the WPRs don't match the sonics for these large and small values of at the end of Section 5. Perhaps this could be expanded and included in the Conclusions?) (2) We have changed the fractional biases to be ratios, rather than percentages, as requested. As to the final comment, we do not have an explanation why this behavior exists, and therefore don't include any conjectures. (3) Updated (new) Figs 4c and 7c, as well as (new) Figs 10 and 11.

(1) Figs 12 and 14 would also benefit from additional detail. I'd like to see R2 values for the plots in Fig 12. They look like blobs. In Fig 14, a subplot of residuals would give a more accurate view of the differences between the dissipation rates measured by the sonics versus those measured by the WPRs. By eye, the plots look close, but the residuals might show otherwise. (2) We've added the statistics onto the scatterplots. As for Fig 14, the large range of turbulence values would require plotting the residuals on log axes also, adding no more information than the time series. The scatter plots and box plots are used to show the variations in log-log format.

I also have a short list of secondary scientific comments. Line numbers are given where appropriate. (1) 71: It would be helpful to move the description of  $\Delta R$  to Eqn 11 (I had to flip back to find it) (2) We have included a repeated description here as well. (3) line 164: "pattern and  $\Delta R$  is the height of each range gate."

(1) In Eqn 8,  $\varphi E(k)$  isn't defined; is this the 3D spectrum? (2) It is defined in Eq 3, but was missing the second subscript specifying the direction. It has been added: (3) Eqn 8:  $\varphi E_i(k)$

(1) In Figure 1, it's difficult to distinguish the pale and bright colors (2) We've updated the figure to include circles on one set of lines, and dash the other, to better distinguish the two, even when virtually equal. (3) (new) Fig. 2 caption: "...with (bright colors with circles) and without (pale colors, dashed lines) the adjustments..."

(1) 135: I suggest "interval" instead of "inertial range", since you can't be sure that this is the inertial range (2) Instead, we state that it is an assumption that this interval is the
inertial range: (3) line 138: "...inertial range (here, identified by visual inspection to be from..."

(1) 136: I would prefer a more authoritative reference than the one given (2) We've replaced it with the more definitive Kaimal reference (3) line 139: "as in Kaimal et al. (1968), to solve for  $\varepsilon$ :..."

(1) In Figs 9 and 10, the purple is really close to the red; could a different color be used? (2) We've changed it to green, to stand out more. (3) Updated (new) Figs 10 and 11.

(1) 392: Strike "very" (2) Fixed. (3) line 404: "...compared well..."

3 Technical corrections There are several run-on and clumsily structured sentences. One more read-through by the authors, which will likely come about through the review process, will undoubtedly help. (1) 21 "Wind profiling radars...": Run-on sentence, with "introduced" used twice (2) Fixed. (3) line 21: "Wind profiling radars (WPRs) showed the possibility of observing profiles of turbulence in the planetary boundary layer (PBL) in Hocking (1985), wherein turbulence intensities and dissipation rates were observed using backscatter intensities and Doppler spectral widths."

(1) 41 "A 300-meter...": Run-on sentence (2) Fixed. (3) line 43: "...located at the site. During the..."

(1) 44: "6" should be "six" (2) Fixed. (3) line 45: "six"

(1) 45: "wind profiling radars" should be "WPRs" (2) Fixed. (3) line 46: "WPRs"

(1) 46: Strike "the" after "BAO" (2) Fixed. (3) line 47: "the BAO Visitors' Center"

(1) 76 "If desired...": Run-on sentence (2) This paragraph has been re-written, adjusting this sentence as well. (3) paragraph starting at line 65

(1) In Eqn 3, the subscript i is used to represent velocity components, while on line 121, xx is used (2) Fixed. (3) line 123 and Eq 9: "Tii"
(1) In Eqns 4, 6, and 11, three different symbols are used for mean wind speed (2) Fixed. (3) line 161 and Eq. 12: uses over-bar for mean

(1) 114: The end of the sentence might read better as "which operate at sufficiently high frequencies to resolve the inertial range" (2) Fixed. (3) line 119: "which operate at sufficiently high frequencies to resolve the inertial range"

(1) 120: Strike "used to move from frequency to wavenumber space" (2) Fixed (3) line 122: "...Taylor's hypothesis, Wyngaard..."

(1) 133: Strike "It is seen that" (2) Fixed. (3) line 136: "...adjustments. That..."

(1) 151: Run-on sentence (2) Fixed. (3) line 162: "...properties. For a vertically-pointing beam, the contribution is determined..."

(1) 164, 255: Use "percent" instead of symbol (2) The second reviewer requested these biases in ratios, rather than percentages, so this has been removed in line 255 (3) line 174: "40 percent line 264: "fractional bias equal to 1"

(1) 175: Sentence might read better as "The integral in Equation 12 can be solved by converting to spherical coordinates" (2) Fixed. (3) line 184: "The integral in Equation 13 can be solved by converting to spherical coordinates:"

---

## Author Comment (AC2) · 12 Jan 2017

We thank the reviewer for their detailed review, for which they obviously put in a lot of time and effort. Although the review was quite negative, we believe that much of the criticism stemmed from one or two fundamental misunderstandings that the reviewer had, in part due to our poor choice of a single word or a sentence. The first of these misunderstanding was due to our use of the word "adjacent" in describing the location of the WPR's relative to the tower. Although we later quantified that "adjacent" meant they were separated by 600m, the reviewer apparently missed this point, and believe that they were in fact co-located to within a few meters or tens of meters. Based on that assumption, the reviewer then asked for dissipation calculations and inter-comparisons on time scales of seconds, allowing for inclusion of the intermittency of the turbulence. With a 600m spatial separation comparisons of dissipation rates on these short time

scales would make no physical sense. Our goal, which we have now also re-iterated and made clearer in the manuscript, was not to study the fine-scale structure of dissipation in the atmosphere, but to calculate O(1 hr) dissipation rates that can be used to evaluate turbulence parameterization schemes in NWP models. The second apparent misunderstanding is how the dissipation rates are calculated from WPR data. The reviewer apparently misunderstood that these were being calculated from ultra-high (20 Hz) radar measurements of the radar velocity time series, much in the same way that they are calculated from sonic anemometer velocity time series. Hence the reviewer's question about units of Hz in the width of the radar Doppler spectrum. The WPR dissipation rates are in fact calculated from the radar Doppler spectrum that has units of Power versus velocity, not power versus frequency. We now include an example of a radar Doppler spectrum to help make this point, and we also include a clearer description of the difference between the two. Finally, the reviewer criticizes the manuscript for what they consider to be still a low correlation between the WPR and sonic anemometer dissipation rates. We certainly agree that the results are far from perfect. However, science is incremental by nature, and our results are undeniably a meaningful improvement over previously published results comparing the two. Given the basic misunderstandings of the reviewer, we have clarified our language and descriptions, taken the reviewer's other constructive comments into consideration (see below), and have decided to resubmit a revised manuscript. We hope that the reviewer will now see the merit in this publication.

(1) Referee Comment (2) Author's Response (3) Manuscript Change

General comments: (1) The authors present and discuss observations of the turbulent kinetic energy dissipation rate $\varepsilon$ retrieved from collocated sonic and clear-air wind profiling radar (WPRs) measurements. While the measurements were collected with state-of-the-art instrumentation, the authors appear to not have taken full advantage of the potential of their data. In particular, the authors rely on spectra estimated from unnecessarily long time series, thereby averaging out most of the $\varepsilon$ variability that is

characteristic of the intermittent atmospheric boundary layer. Moreover, the authors do not present any raw data (sonic time series, WPR signal time series) or any conclusive intermediate steps in the data processing, such as Doppler spectra and dwell-time-by-dwell-time time series of spectral-width estimates. So there is no way to find out why the $\varepsilon$ correlograms (figures 4, 5, and 7) show such excessive discrepancies between the sonic retrievals and the WPR retrievals of $\varepsilon$ (about three orders of magnitude). (2) We have recognized a misunderstanding by the reviewer, which we hope we can clear up with the inclusion of more detail in our introduction and a figure of a raw Doppler spectrum. The motivation of this work is not to measure the small-scale intermittency of the boundary layer, but to determine dissipation rates on time scales appropriate for evaluation of NWP models, O(15-60min). We also would like to draw attention to the attached manuscript on measuring velocity variance from WPRs, which we expect to be accepted for publication in AMT very soon (very few, minor revisions were requested and have been submitted). That manuscript includes more intermediate steps, which may clear up other misunderstandings on the method here, and contain sufficiently significant results to warrant separate publication. (3) New Figure 1 line 34: "With these measurements, comparisons can be made on time scales appropriate for evaluation of numerical weather prediction models, O(1hr)." line 74: "Fast-Fourier Transformations (FFTs) were then computed to obtain Doppler spectra (power versus velocity), as shown in Fig. 1 from the 449-MHz WPR..."

(1) The abstract does not contain any quantitative and hard information that would substantiate the claim in the title ("Improved observations ..."). The introduction lists a number of relevant articles but says almost nothing about the underlying physics. I would have expected the introduction to contain a critical discussion of the strengths and weaknesses of the various techniques to retrieve energy dissipation rates from high-resolution point measurements (sonics) and from radar wind profiler measurements. (Note that the paper does not even have a dedicated discussion section.) (2) We wished to keep the focus of this paper on furthering the historical work of measuring dissipation rates with WPR, rather than delving into the physics behind the methods.

The use of sonic anemometers is well established, and therefore we believe it requires no further support here, and in any case, this would be beyond the scope of this paper. Interested readers can use our many listed references for more background, if desired, but our application at hand is the improvement of a method for comparison with models, not the theoretical physics of turbulence.

(1) I cannot recommend this paper for publication. The following specific comments may help the authors to rethink their approach and to re-process the data. (2) We hope that our responses below are satisfactory for addressing the reviewers concerns. We've recognized that we were unclear, and lead to a major misunderstanding of our method, so we are grateful for these comments so we can be sure future readers will not be confused.

Specific comments: (1) 1. 5f.: ". . . two WPRs operated in an optimized configuration, using high spectral resolution for increased accuracy of Doppler spectral width . . ." — The spectral resolution of a Doppler spectrum is determined by the dwell time (and not, as one might erroneously believe, by the pulse repetition period or by the sampling rate or by the coherent integration time). So a high spectral resolution requires a long dwell time. On the other hand, the dwell time should not be excessively long because the dwell time determines also the achievable time resolution. Because the energy dissipation rate $\varepsilon$ in the atmospheric boundary layer is a highly intermittent quantity, one might want to limit the dwell time such that "coherent structures" can be captured. So in what sense is the WRP operation mode optimized here? What is the "optimum" compromise between good spectral resolution and good time resolution? (2) Optimized turbulence mode has a large number of FFT points for higher resolution of the Doppler velocity spectrum, while keeping a short dwell time, for flexibility in determining the optimal number of spectral averages. We realize we may have been unclear as to what the Doppler spectrum is, which is then used for the calculation of the spectral width and dissipation rate. It is not the retrieved backscatter amplitudes and phases, but the Fourier transform of that time series, which is a function of velocity. This provides the

distribution of velocities of backscattering particles within the radar volume, the spread of which indicates the level of turbulence. We hope this clarifies many of the questions raised in subsequent comments. (3) the paragraph surrounding, and including line 148: "The Doppler spectrum provides the distribution of velocities of backscattering particles within the radar volume, with an average which equal to the mean velocity, and a spread which indicates the level of turbulence."

(1) 12f.: "Resulting measurements of turbulence dissipation rates correlated well (R2 = 0.57) with sonic anemometers . . ." — Is this the correlation coefficient between WRP- and sonic- retrieved estimates of $\varepsilon$, or the correlation coefficient between WRP- and sonic-retrieved estimates of the logarithm of $\varepsilon$? The figures 4, 5, and 7 appear to indicate the latter. A moderate correlation coefficient between the logarithms of $\varepsilon$-estimates that vary over five orders of magnitude does not speak in favor of the quality of the retrievals. Please explain. (2) This is the correlation between the logarithms of $\varepsilon$. Considering the range over several orders of magnitude, the correlation between $\varepsilon$ greatly skews the large values. This is part of how this study improved over previous measurements, where the small values were vastly over-estimated, but that impact was not seen in linear units.

(1) 21f.: "Wind profiling radars (WPRs) introduced the possibility of observing full profiles of turbulence in the planetary boundary layer . . ." — What do you mean by "full profiles of turbulence"? First, what is a "full" profile? Second, turbulence is characterized by many different parameters, such as the energy dissipation rate, the temperature structure param- eter, the inner scale, the temperature variance dissipation rate, the sensible heat flux, the vertical-velocity variance, etc. So what do you mean with "turbulence"? (2) First, "full profiles" referred to the freedom from the restraints of the height and placement of point measurements on a tower. Realizing this ambiguity, we've removed this word. Second, the rest of the sentence includes the mention of multiple measurements of turbulence (turbulence intensity and dissipation rate), and the following sentence discussed the identification by Cohn 1995 of dissipation rate

as the most promising metric. (3) line 21: "Wind profiling radars (WPRs) showed the possibility of observing profiles of turbulence in the planetary boundary layer (PBL) in Hocking (1985), wherein turbulence intensities and dissipation rates were observed using backscatter intensities and Doppler spectral widths." (1) 90-95: The TKE budget equation is not relevant for the retrieval of $\varepsilon$ from sonic or WPR measurements discussed in this paper, and therefore should be deleted.

(2) Since we have added the emphasis on using these measurements for model verification, the TKE budget is indeed very relevant as it relates dissipation rates to NWP model parametrization schemes, and we therefore respectfully disagree, and have chosen to keep it.

(1) 104: ". . . where $\alpha i$ is a constant." — Do you mean to say that there are three different Kolmogorov constants, one for each velocity component? This is misleading. There are only two different Kolmogorov constants: one for the longitudinal velocity component (the velocity component parallel to the direction of the wave vector) and one for the transverse (lateral) velocity component. The underlying physics can be found in Monin and Yaglom (1975, p. 355).

(2) Since the constants are not defined here, the fact that the transverse and vertical component constants are equal was not discussed. We've added a note, as well as pointed out that, since the velocities have been realigned, that u is the longitudinal direction. Alpha_u is defined where it is used. (3) line 104: "velocity component, i=[u,v,w]" line 109: "where alpha_i is a constant (and alpha_v=alpha_w)" line 127: "subscripts i=u, v, and w denote the components of the velocity vector (longitudinal, transverse, and vertical, respectively)."

(1) 115: "WPRs do not collect data at a high enough frequency to observe the inertial range, and therefore must rely on spectral widths for their small-scale variance measurements . . ." — According to Table 1, the WPRs' interpulse periods are 33 and 45 $\mu$s. Because no coherent integration is performed, this means that the sampling

period of the complex WPR signal is higher than 20 kHz, about 1000 times the sonics' sampling rate. So it is definitely not the sampling rate that matters here. Please explain!

(2) This sentence has been removed, because it was out of place here. However, we recognize a misunderstanding of our method, in that the sampling of the velocity time series is set by the dwell time, which is the product of the inter-pulse period, the number of coherent integrations, the number of FFT points, and the number of spectral averages. The resolution of the time series of complex amplitudes and phases is not the resolution relevant to the Doppler spectral widths, and therefore dissipation rates.

(1) 117f. "Using the high-sampling frequency measurements by the sonic anemometers, the vari- ance observed can be directly obtained from the energy spectrum for the dissipation rate." — It is not the "variance observed" that is used for the retrieval of $\varepsilon$. Please clarify! (2) We've clarified: (3) line 119: "This can be accomplished using in situ sonic anemometers, which operate at sufficiently high frequencies to resolve the inertial range. Using the high-sampling frequency measurements by the sonic anemometers, the dissipation rate can be directly obtained from the energy spectrum"

(1) 138: "Spectra were computed for each 15-minute interval, with corresponding dissipation rates. The dissipation rates were then averaged over 30 mins to compare with those obtained from the WPRs." — Why do you average over such long periods? It should be possible to get clean $\varepsilon$ estimates from the sonic data and from the WPR spectral widths for periods of 1 min or even shorter! At a sonic sampling rate of 20 Hz, you have 1200 velocity data points per minute, which is more than enough to provide a clean estimate for $\varepsilon$. The same should be the case for $\varepsilon$-estimates retrieved from the Doppler spectral widths obtained from 1-min long WPR signal time series.

(2) The dissipation rates from the Doppler spectral width from WPRs are retrievable only at the time scales of the dwell time (defined above, and now included in Eq. 1). Furthermore, considering the motivation of model comparisons, longer time scales are

output from models, so 15-30 minutes is more reasonable for this application. Furthermore, at such short timescales (O(1 min)), the 600 m distance between instruments would make comparisons irrelevant, as the assumption of homogeneity is not accurate over those length and time scales. (3) paragraph starting at line 65: "The Doppler spectra are retrieved at a temporal resolution, or a dwell time, $\Delta$t, determined by the radar set-up parameters: $\Delta$t = [IPP][NCOH][NFFT][NSPEC], where IPP is the inter-pulse period (ns), NCOH is the number of coherent integrations, NFFT is the number of points used in the fast Fourier transform, and NSPEC is the number of spectral averages."

(1) "The spectral width of the Doppler spectrum is twice the standard deviation, $\sigma$m, of the unresolved velocities in the measurement volume during each dwell." — Whether or not this statement is correct depends on what exactly you mean by "unresolved velocities in the measurement volume." Moreover, the width of the Doppler spectrum has the unit Hz while the unit of $\sigma$m is ms$-1$, so the ratio between spectral width and $\sigma$m is not a dimensionless number. Please clarify! (2) The resolved velocities are those captured in the time series of velocities (first moments of Doppler spectrum), and the unresolved velocities are all the velocities that make up the Doppler spectrum. The dependent variable in the Doppler spectrum is velocity, not frequency, so has units of m s-1. We hope the previous discussions have clarified this.

(1) 180: "The dissipation rates were estimated for the 30 minutes of turbulence mode, when the backscatter intensity time series were collected . . ." — The dissipation rates are not retrieved from intensity time series but from time series of amplitudes and phases (i.e., from time series of the complex radar signal). Please clarify! (2) By "backscatter intensity," we mean "amplitude and phases," but have changed the language to match. However, the dissipation rates are not retrieved from the time series of amplitudes and phases, but from the width of the Doppler spectrum, which is the FFT of the complex time series. (3) line 188: ". . .when the time series of amplitudes and phases (I and Q) were saved, and after different . . ."

(1) 186: ". . . backscatter intensity time series filtering . . ." — See previous

comment. Please clarify! (2) See previous comment. (3) line 194: "...time series filtering (of amplitude and phase signal), ..."

(1) 291: "Each dwell collected by the 449-MHz WPR spans about 13 seconds (and the 915-MHz, about 17 seconds; see Table 1), capturing only a short period of the atmosphere's motions."   For short dwell times, the Doppler spectral width provides information about the spatial variability of the radial wind velocity vr within the radar's resolution volume. For long dwell times, the spectral width is contaminated by temporal variations of the mean (averaged over the radar's resolution volume) vr. It is the spatial variability of vr within the radar's resolution volume that provides the most direct information of $\varepsilon$. The fact that $\varepsilon$-estimates vary erratically from dwell time to dwell time must not be misinterpreted as instrumental noise that has to be averaged out; more likely, the "noise" represents the intermittent nature of $\varepsilon$ in the high-Reynolds number ABL (Kolmogorov, 1962; Obukhov, 1962). The local energy dissipation rate in high-Reynolds number turbulence (as in the ABL) is approximately lognormally distributed, such that variations of $\varepsilon$ over many orders of magnitude within short time scales (i.e., minutes or less) are to be expected. It would be very interesting to see to what extent the collocated sonic and WPR $\varepsilon$ retrievals track each other at time scales between 10 s and 10 min, rather than at time scales larger than 10 min. (2) We agree with all of the reviewer's comments about turbulence intensity. However, this experiment is not set up to make these types of comparisons, with 600 m between instruments. They cannot be considered "co-located" over intermittency-related time scales. Averaging is used to make more appropriate comparisons with NWP models (as well as to increase the SNR). It would be an interesting study to more directly collocate a WPR with sonics to see how the shorter time scales are captured by the spectral widths of the WPR measurement volume.